# Survey for ‘*Candidatus* Liberibacter’ and ‘*Candidatus* Phytoplasma’ in Citrus in Chile

**DOI:** 10.3390/pathogens11010048

**Published:** 2021-12-31

**Authors:** Nicolas Quiroga, Camila Gamboa, Gabriela Medina, Nicoletta Contaldo, Fernando Torres, Assunta Bertaccini, Alan Zamorano, Nicola Fiore

**Affiliations:** 1Departamento de Sanidad Vegetal, Facultad de Ciencias Agronómicas, University of Chile, Santiago 8820808, Chile; nicolasquirogabarrera@gmail.com (N.Q.); camila.gamboa@ug.uchile.cl (C.G.); gabriela.medina@ug.uchile.cl (G.M.); agezac@u.uchile.cl (A.Z.); 2Programa de Doctorado en Ciencias Silvoagropecuarias y Veterinarias, Campus Sur Universidad de Chile, Santiago 8820808, Chile; 3Department of Agricultural and Food Sciences, Alma Mater Studiorum—Università di Bologna, 40127 Bologna, Italy; nicoletta.contaldo2@unibo.it (N.C.); assunta.bertaccini@unibo.it (A.B.); 4Subdepartamento Vigilancia y Control de Plagas Agrícolas, Departamento Sanidad Vegetal, División Protección Agrícola, Servicio Agrícola y Ganadero, Forestal y Semillas, Santiago 8320000, Chile; Fernando.torres@sag.gob.cl

**Keywords:** citrus diseases, phytoplasmas, Liberibacter, nested-PCR/RLFP, sequencing, molecular identification

## Abstract

The considerable economic losses in citrus associated with ‘*Candidatus* Liberibacter’ and ‘*Candidatus* Phytoplasma’ presence have alerted all producing regions of the world. In Chile, none of these bacteria have been reported in citrus species. During the years 2017 and 2019, 258 samples presenting symptoms similar to those associated with the presence of these bacteria were examined. No detection of ‘*Ca*. Liberibacter’ associated with “huanglongbing” disease was obtained in the tested samples; therefore, this quarantine pest is maintained as absent in Chile. However, 14 plants resulted positive for phytoplasmas enclosed in subgroups 16SrV-A (12 plants) and 16SrXIII-F (2 plants). Although they have been found in other plant species, this is the first report of these phytoplasmas in citrus worldwide.

## 1. Introduction

Different species of ‘*Candidatus* Liberibacter’ have been reportedly associated with “huanglongbing” (HLB), the most important disease in citrus worldwide [1,2]. This disease is spread through a large part of citrus-growing areas worldwide [3,4]. In South America, ‘*Ca*. L. asiaticus’ and ‘*Ca*. L. americanus’ are present in Brazil [5,6] and Paraguay [7], while only ‘*Ca*. L. asiaticus’ is reported in Argentina and Colombia [8,9]. Furthermore, *Diaphorina citri*, the main vector of both bacteria, has been found in all these countries, and also in Uruguay [10], Ecuador [11], and Venezuela [12]. In the American continent, the presence of ‘*Ca*. L. africanus’ and its vector *Trioza erytreae* were never reported. Until recently, HLB-associated bacteria and their insect vector(s) have not been detected in Chile; since 2011, monitoring plans have been developed to prevent the entry of the disease’s agents and their insect vectors [13]. Given Chile’s geographic proximity to many countries in which they are present, the risk of the disease spread into its territory is high. Therefore, the Chilean citrus industry is making a great effort in monitoring and verifying the absence of these pathogens and of their insect vectors.

It has been repeatedly observed that citrus plants infected by ‘*Candidatus* Phytoplasma’ show the typical symptoms observed in HLB infected plants, such as irregular yellowing of the leaves and weakening of the tree. In some cases, it was possible to verify co-infection between the two bacteria [14,15,16]. For this reason, both pathogens were surveyed to verify the local situation [17] with a meticulous inspection followed by specific molecular tests to verify the presence of these bacteria in citrus plants in Chile. The sampling was mainly directed to plants showing symptoms attributable to these bacteria in all the citrus growing areas of the country.

## 2. Results

All the PCR analyses for the detection of ‘*Candidatus* Liberibacter’ species in citrus samples were negative. However, 14 samples were positive for phytoplasma presence after amplification on 16S rRNA and *Ssu12p* genes [18,19]. The amplicons obtained were cloned, and five clones of each sample were sequenced. The sequences resulting from the five clones obtained from each sample were identical and were aligned and compared with the phytoplasma sequences available in GenBank (NCBI).

A 100% nucleotide identity in the 16S rRNA gene was present with samples enclosed in the ‘*Ca*. P. ulmi’ and ‘*Ca*. P. hispanicum’ strains. In the case of the *Ssu12p* gene, the nucleotide identity percentages observed were 99.52% and 100% in comparison with the sequences available in GenBank, respectively. Among the 14 phytoplasma-positive samples, 12 were infected by ‘*Ca*. P. ulmi’ (elm yellows group, 16SrV-A), and two were infected by ‘*Ca*. P. hispanicum’ strains (Mexican periwinkle virescence group, 16SrXIII-A). Maximum parsimony phylogenetic reconstruction using 16S rRNA and *Ssu12p* gene sequences, including the strains obtained in this study, confirm that the detected phytoplasmas cluster with phytoplasmas enclosed in ribosomal groups 16SrV and 16SrXIII. In particular, phytoplasma strains from the samples CTC 202, CTC 199, and CTC 192 are closely related to the strain EY1 (GenBank accession number AY197655) (Figure 1A). Strains CTC 170 and CTC 134 grouped with those of the ribosomal subgroup 16SrXIII-F, in particular, strain StrCL-1 (GenBank accession number MH939193), from a Chilean strawberry sample. The phylogenetic tree of the *Ssu12p* gene confirms these phylogenetic relationships (Figure 1B).

Furthermore, the sequence of sample CTC 192 (deposited in GenBank under the accession number OL677628) was subjected to RFLP in silico to complete the identification of the phytoplasmas at the ribosomal subgroup level using the enzymes *Rsa*I and *Bfa*I for phytoplasmas enclosed in the 16SrV group [20]. The virtual digestion with the *Rsa*I enzyme showed that the profiles of the strains corresponding to groups 16SrV-A, 16SrV-C, 16SrV-D, and 16SrV-E were identical to the one of this strain. The digestion with the *Bfa*I enzyme generated differentiable profiles with the strains of the mentioned subgroups, identical only to those of the ribosomal subgroup 16SrV-A (Figure 2). The enzymes used for in silico RFLP of the phytoplasma strain CTC170 (deposited in GenBank under the accession number OL672243) were *Kpn*I and *Rsa*I [21]. Both digestion profiles show that the Chilean strains are identical to the strains in the subgroup 16SrXIII-F (Figure 3).

The molecular characterization of the strains from 14 citrus samples indicates that 12 samples were positive for phytoplasmas enclosed in the subgroup 16SrV-A, and two samples were positive for phytoplasmas enclosed in subgroup 16SrXIII-F. The overall percentage of phytoplasma infection in the plants was 5.43%. The citrus samples positive for 16SrV-A phytoplasmas were collected in two orchards, both located in the Metropolitana Region (Table 2). The symptoms showed by these plants were leaf yellowing undistinguishable from those reported as associated with the presence of ‘*Ca*. Liberibacter’ (Figure 4). The samples positive for 16SrXIII-F phytoplasmas were collected in two orchards located in the O’Higgins Region (Table 2), showing (sample CTC134) symptoms of generalized yellowing of the tree, abortion of fruits, defoliation of shoots, and asynchronous flowering compared with non-infected trees. Sample CTC 134 showed threadlike leaves and witches’ broom shoots (Figure 5).

## 3. Discussion

This survey provides results about the monitoring of HLB and phytoplasma presence in citrus species in Chile. The country maintains the absence of bacterial species and insect vectors of HLB disease in citrus plants. It is important to note that the Chilean Agricultural and Livestock Service (SAG) has, since 2011, been carrying out an intense sampling monitoring to detect these bacteria and their main insect vector, *D. citri*. In addition, SAG personnel are constantly developing technical dissemination and training sessions for producers with the objective of preventing a possible epidemic with timely detection, avoiding the spread of the disease in Chilean territory. To date, 7809 samples from all citrus-producing regions in Chile were tested by SAG (personal communication), including citrus plants used as ornamentals [22]. In this study, highly suspected plants were sampled, and different detection protocols were applied in different laboratories in and out of Chile to confirm the results obtained.

Symptoms attributable to HLB have been observed in various citrus orchards in Chile. However, molecular tests to verify the liberibacters presence were negative. It is possible to attribute the symptom observed to various factors, such as nutritional deficiencies, insect attack, presence of soil pathogens, or insufficient general agronomic management. This work confirms that phytoplasmas are associated with symptoms like those induced by ‘*Ca*. Liberibacter’ species in citrus, as reported in China, Brazil, Mexico, and the Caribbean (Cuba, Guadeloupe, and Jamaica), where phytoplasmas were also identified in plants with HLB symptoms and co-infections between pathogens [14,15,17,23,24]. Moreover, the only phytoplasma presence was associated with HLB-like symptoms in both Asian and American countries [23,24,25,26,27]. The samples positive for phytoplasma 16SrV-A in this study showed leaf symptoms resembling those shown by HLB-infected plants. This situation has been frequently observed in regions where other phytoplasmas have been identified as infecting citrus [14,15,17,28]. On the other hand, citrus samples positive for phytoplasma 16SrXIII-F showed symptoms more commonly associated with the presence of these pathogens, such as witches’ broom, severe leaf deformation, and flower alterations. This symptomatology has also been described in citrus plants infected by 16SrII group phytoplasmas (‘*Ca*. P. aurantifolia’ and related strains) in Asia and Australia [27,29].

The information provided by this study represents the first report of phytoplasmas in citrus plants in Chile. Furthermore, this is the first detection of the phytoplasmas 16SrV-A and 16SrXIII-F in citrus plants in the world. Phytoplasmas of the ribosomal group 16SrII have been reported to infect citrus plants in Brazil, China, and Iran [22,30,31]. Furthermore, in Brazil, the phytoplasmas of the ribosomal groups 16SrIII and 16SrIX have been detected [17,28]. In Mexico and China, a ‘*Ca*. P. asteris’ strain (16SrI) was reported [14,15]. Specific detections of ribosomal subgroups not widely distributed worldwide include the 16SrIX group in lemon and orange trees in Puerto Rico [32] and a phytoplasma of the 16SrXIV-A subgroup in lemon trees in India [33]. The presence of 16SrV-A phytoplasma in Chile was previously identified in vineyards of the Metropolitana Region, associated with reddening and short internodes of the grapevine shoots [34]. Moreover, transmission tests with the leafhopper *Amplicephalus curtulus*, widespread in the central zone of Chile, showed the ability of this insect species to transmit the 16SrV-A phytoplasmas [35]. On the other hand, the phytoplasma 16SrXIII-F was detected for the first time in Chile in strawberry orchards of the Region of Valparaíso, associated with symptoms of phyllody and virescence [36]. Subsequent studies showed that this phytoplasma was present in all the strawberry production regions in Chile, including the localities where positive citrus samples have been detected in this survey [21,37]. In addition, this phytoplasma has been reported in calafate (*Berberis microphylla* G. Forst.), a shrub of the Berberidaceae family native to Chilean and Argentinian Patagonia [38,39]. So far, the vector(s) of this phytoplasma is not known. The 16SrXIII group is widely distributed in the other countries of South America, affecting various crops such as potato (*Solanum tuberosum* L.), broccoli (*Brassica oleracea* L.), and papaya (*Carica papaya* L.), among others [40,41,42,43].

Evidence shows that citrus phytoplasma infections are sporadic in isolated orchards, sometimes in remote areas surrounded by spontaneous vegetation. In times with higher-than-average rainfall during spring and autumn, the growth of spontaneous shrubs and weeds is favored, thereby increasing insect-feeding in these plants. Insect vectors can, therefore, sporadically visit crops and transmit phytoplasmas, suggesting that spontaneous vegetation may also be a reservoir for these phytoplasmas [29,44,45]. This situation has been evidenced in epidemiological studies associated with grapevine yellows in Chile [46,47,48].

The presence of viruses and viroids that infect citrus plants in Chile could lead to confusion among the symptoms. It is essential to consider all these factors in the climate change scenario. Studies carried out in Chile show that the climatic conditions that are projected for the future could influence the appearance of diseases of alien origin [49]. The discovery of phytoplasmas associated with symptoms like those of HLB reinforces the need to continue monitoring and promoting epidemiological studies on citrus. Preventing these diseases is everyone’s job, including governments, nurseries, technical advisers, producers, and researchers. Chile is surrounded by countries in which HLB and its insect vector *D. citri* are present. Fortunately, the country has climatic and geographical barriers that have contributed positively to the present local phytosanitary situation. Chile has great potential as a citrus-growing country and being an HLB-free territory is a commercial advantage that should not be lost.

## 4. Materials and Methods

From the main citrus-producing areas of Chile, 258 samples were collected in two areas during two periods. From December 2017 to April 2018, the regions of Tarapacá, Coquimbo, and Valparaíso were surveyed. The second sampling period was carried out between November 2018 to April 2019 in the Regions of Libertador Bernardo O’Higgins, Maule, and Metropolitana. The number of samples was established based on the number of hectares planted per region; one sample was collected for every 100 ha (Table 3). The samples were collected mainly from citrus trees that showed symptoms referable to HLB, like yellow shoots, leaves with asymmetric chlorotic spots, and thick and leathery texture [50]. In addition, trees with symptoms associated with phytoplasma presence, such as witches’ broom, flower abnormalities, and general decline, were sampled [15,27,29]. Samples that presented symptoms of water deficiency, nutritional deficiencies, nematode, and insect attacks, and phytotoxicity were not collected. In orchards where no symptoms attributable to these pathogens were observed, asymptomatic plants were randomly sampled. Each sample corresponds to one tree, from which four 15 cm long shoots with at least 5 leaves were collected from different sides of the tree. The geographic coordinates and the origin of the collected samples were recorded. The samples were transported in thermoregulated containers and stored at 4 °C before nucleic acid extraction.

DNA was extracted from 1 g of midribs using the CTAB method [48]. The nucleic acids were dissolved in Tris-EDTA buffer pH 8.0 and kept at 4 °C. All samples were analyzed by PCR and nested-PCR. For the detection of ‘*Ca*. Liberibacter’ species five amplification protocols were used: three universal 16S and 23S gene PCRs [51,52]; a PCR-duplex, for the simultaneous and specific detection of the three HLB-associated ‘*Ca*. Liberibacter’ species [5,53], and a universal nested PCR for the three species [54,55]. Phytoplasma detection was performed by nested-PCR of the 16S rRNA gene [18] and the gene coding the ribosomal *Ssu12p* gene [19]. The PCR products obtained for both genes were purified from the agarose gel using the EZNA^®^ Gel Extraction Kit (Omega Bio-tek, Norcross, GA, USA). The DNA fragments were ligated into the cloning vector pGEM-T Easy, following the manufacturer’s instructions (Promega Inc., Fitchburg, WI, USA). Putative recombinant clones were analyzed by colony PCR and selected fragments sequenced in both directions at Psomagen Inc. (Rockville, MD, USA). The sequences were aligned with the GenBank database using the BLAST engine for local alignment (Blast version N 2.2.12) and compared with those of phytoplasmas published in the National Center for Biotechnology Information (NCBI) available on the internet (http://www.ncbi.nlm.nih.gov/blast/, accessed on 15 November 2021) [56]. The restriction fragment length polymorphism (RFLP) analysis was performed in silico with the appropriate restriction enzymes according to the ribosomal groups obtained. The in silico RFLPs were generated in the *i*PhyClassifier online tool (https://plantpathology.ba.ars.usda.gov/cgi-bin/resource/iphyclassifier.cgi, accessed on 15 November 2021) [57].

## 5. Conclusions

The results presented in this study indicate that Chile is free from the “huanglongbing” associated bacteria, maintaining its status of absence of this quarantine pathogen. In the citrus industry of Chile, government services and researchers maintain a constant interaction to prevent the entry of these pathogens and their insect vectors. In some plants that presented symptoms like those of HLB, phytoplasmas enclosed in 16SrV-A and 16SrXIII-F subgroups were detected. This is the first report of phytoplasmas in citrus in Chile; the identified phytoplasmas were not described in other citrus-growing regions in the world.

## Figures and Tables

**Figure 1 pathogens-11-00048-f001:**
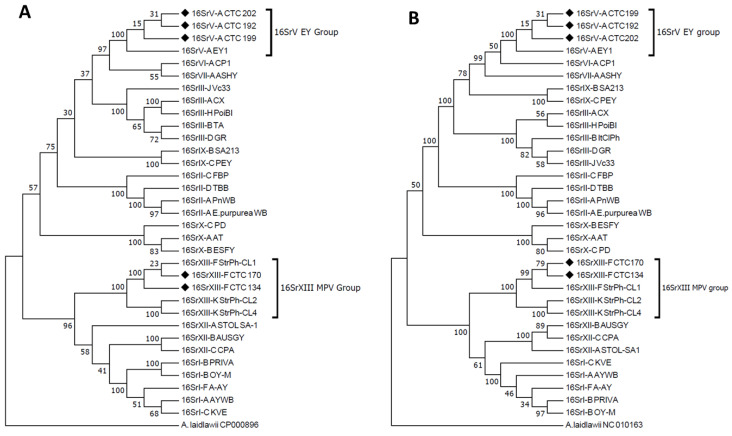
Phylogenetic tree of (**A**) 16S rRNA gene region (1240 nt) and (**B**) *Ssu12p* region (724 nt) enclosing phytoplasmas detected in citrus from Chile, highlighted with filled diamonds, and selected ‘*Ca*. Phytoplasma’ strains. Information on the phytoplasma strains is reported in Table 1. The tree was constructed using the maximum parsimony algorithm. The numbers in the nodes represent starting values based on 500 pseudo-replications for stability estimation and clade support *A. laidlawii* is used as outgroup strain.

**Figure 2 pathogens-11-00048-f002:**
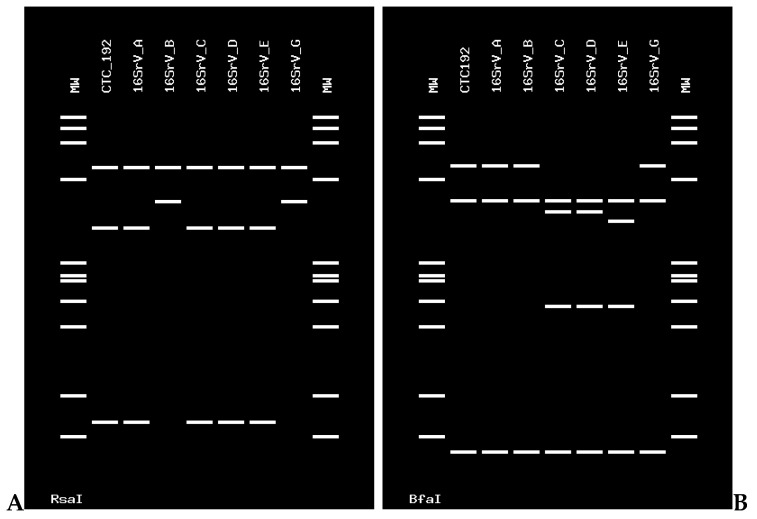
In silico RFLP profiles of the phytoplasma strains in the 16SrV subgroups. (**A**) Restriction profiles generated by the *Rsa*I enzyme. (**B**) Restriction profiles generated by the *Bfa*I enzyme. CTC 192 is a citrus sample. MW: molecular marker PhiX174 digested with *HaeII*I. Fragment size (nt) from top to bottom: 1353, 1078, 872, 603, 310, 281, 271, 234, 194, 118, and 72. Phytoplasmas used for comparison are: 16SrV-A, elm yellows (EY) ‘*Ca*. P. ulmi ‘(GenBank accession number: AY197655); 16SrV-B, jujube witches’ broom (JWB-G1) ‘*Ca*. P. ziziphi ‘(GenBank accession number: AB052876); 16SrV-C, “flavescence dorée” (FD-C) (GenBank accession number: X76560); 16SrV-D, “flavescence dorée” (FD-D) (GenBank accession number: AJ548787); 16SrV-E, rubus stunt (RuS) ‘*Ca*. P. rubi ‘(GenBank accession number: AY197648); 16SrV-G, Korean jujube witches’ broom (GenBank accession number: AB052879).

**Figure 3 pathogens-11-00048-f003:**
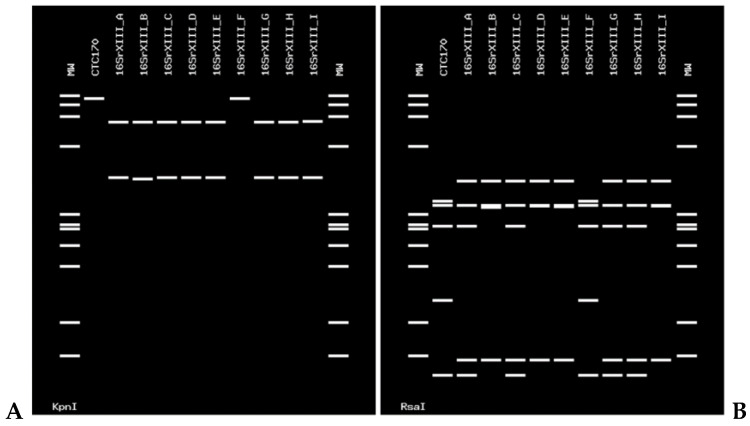
In silico RFLP profiles of phytoplasmas in the subgroups of 16SrXIII ribosomal group. (**A**) Restriction profiles generated by the *Kpn*I enzyme. (**B**) Restriction profiles generated by the *Rsa*I enzyme. CTC 192 citrus sample. MW: molecular marker PhiX174 digested with *Hae*III. Fragment size (nt) from top to bottom: 1353, 1078, 872, 603, 310, 281, 271, 234, 194, 118, and 72. Phytoplasmas used for comparison: 16SrXIII-A, Mexican periwinkle virescence (MPV) ‘*Ca*. P. hispanicum’ (GenBank accession number: AF248960); 16SrXIII-B, strawberry green petal (STRAWB2) (GenBank accession number: U96616); 16SrXIII-C, Chinaberry yellows (CBY1) (GenBank accession number: AF495882); 16SrXIII-D, Mexican potato purple top (SINPV) (GenBank accession number: FJ914647); 16SrXIII-E, papaya apical curl necrosis (PACN) (GenBank accession number: EU719111); 16SrXIII-F strawberry red leaf (GenBank accession number: KJ921641); 16SrXIII-G, Chinaberry yellowing (ChTY) ‘*Ca*. P. meliae ‘(GenBank accession number: KU850940); 16SrXIII-H broccoli stunt phytoplasma (GenBank accession number: JX626329); 16SrXIII-I Mexican periwinkle virescence phytoplasma (GenBank accession number: KT444664).

**Figure 4 pathogens-11-00048-f004:**
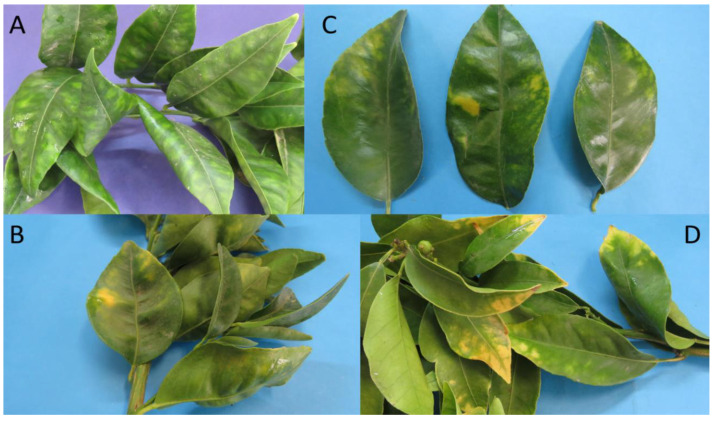
Symptoms of leaf yellowing in the citrus samples resulted positive for 16SrV-A phytoplasmas. (**A**) Orange CTC 192. (**B**) Lemon CTC 199. (**C**) Mandarin CTC 200. (**D**) Orange CTC 202.

**Figure 5 pathogens-11-00048-f005:**
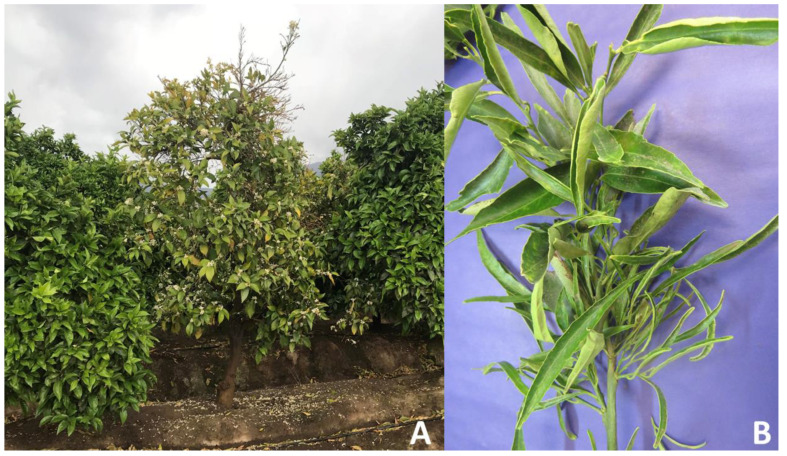
Symptoms in citrus associated with the 16SrXIII-F phytoplasma presence. (**A**) CTC 170, orange plant with generalized yellowing and untimely flowering. (**B**) CTC 134, mandarin plant with threadlike leaves and witches’ broom.

**Table 1 pathogens-11-00048-t001:** List of phytoplasma strains used for the phylogenetic analyses.

16Sr Group	Subgroup	Associated Disease (‘*Ca*. Phytoplasma’ Species)	Acronym	GenBank Accession Number	
				*SSU12p*	16S rRNA
16SrI	A	Aster yellows witches’ broom	AYWB	CP000061	
	B	Onion yellows mild strain	OY-M	NC_005303	
	B	Primrose virescence	PRIVA	MT161512	AY265210
	C	Clover phyllody	KVE	MT161514	AY265217
	F	Aster yellows from apricot	A-AY	MT161515	AY265211
16SrII	A	Peanut witches’ broom	PnWB	AMWZ00000000	
	A	*Echinacea purpurea* witches’ broom	*E. purpurea* WB	LKAC00000000	
	C	Faba bean phyllody	FBP	MT161516	EF193354
	D	Tomato big bud	TBB	MT161517	EF193359
16SrIII	A	Peach × disease (‘*Ca*. P. pruni’)	CX	LHCF00000000	
	B	Italian clover phyllody	ItClPh	AKIM00000000	
	D	Goldenrod yellows	GR	MT161522	FJ376627
	H	Poinsettia branch inducing	PoiBI	AKIK00000000	
	J	Phytoplasma Vc33	Vc33	LLKK00000000	
16SrV	A	Elm yellows (‘*Ca*. P. ulmi’)	EY	MT161527	AY197655
16SrVI	A	Clover proliferation	CP1	MT161528	HQ589189
16SrVII	A	Ash yellows (‘*Ca*. P. fraxini’)	ASHY	MT161529	HQ589190
16SrIX	B	Almond witches’ broom (‘*Ca*. P. phoenicium’)	SA213	JPSQ00000000	
	C	*Picris echioides* yellows	PEY	MT161530	JQ868441
16SrX	A	Apple proliferation (‘*Ca*. P. mali’)	AT	CU469464	
	B	European stone fruit yellows (‘*Ca*. P. prunorum’)	ESFY	MT161533	AM933142
	C	Pear decline (‘*Ca*. P. pyri’)	PD	MT161535	AJ542543
16SrXII	A	“stolbur” (‘*Ca*. P. solani’)	STOL SA-1	MPBG00000000	
	B	Austral. grapev. yellows (‘*Ca*. P. australiense’)	AUSGY	AM422018	
	C	Strawberry lethal yellows	CPA	CP002548	
16SrXIII	F	*Fragaria* × *ananassa* phyllody	StrPh-CL1	MT161538	MH939191
	K	*Fragaria* × *ananassa* phyllody	StrPh-CL2	MT161539	MH939192
	K	*Fragaria* × *ananassa* phyllody	StrPh-CL4	MT161541	MH939194
16SrV	A	*Citrus* × *sinensis* Lane late yellows	CTC192	OL690419	OL677628
16SrXIII	F	*Citrus* × *sinensis* Fukumoto witches’ broom	CTC170	OL690418	OL672243

**Table 2 pathogens-11-00048-t002:** Detail of citrus samples positive to phytoplasmas.

Sample	Species and Variety	Region and Locality	Phytoplasma Detected (Ribosomal Subgroup)
CTC 182	*Citrus reticulata* Murcott	Metropolitana (Pomaire)	16SrV-A
CTC 184	*Citrus reticulata* Murcott	Metropolitana (Pomaire)	16SrV-A
CTC 188	*Citrus reticulata* Murcott	Metropolitana (Pomaire)	16SrV-A
CTC 190	*Citrus* × *limon* Fino 49	Metropolitana (Pomaire)	16SrV-A
CTC 192	*Citrus* × *sinensis* Lane late	Metropolitana (Pomaire)	16SrV-A
CTC 193	*Citrus* × *sinensis* Lane late	Metropolitana (Pomaire)	16SrV-A
CTC 199	*Citrus* × *limon* Eureka	Metropolitana (Mallarauco)	16SrV-A
CTC 200	*Citrus reticulata* Murcott	Metropolitana (Mallarauco)	16SrV-A
CTC 202	*Citrus* × *sinensis* Valencia	Metropolitana (Mallarauco)	16SrV-A
CTC 203	*Citrus* × *limon* Eureka	Metropolitana (Mallarauco)	16SrV-A
CTC 207	*Citrus reticulata* Murcott	Metropolitana (Mallarauco)	16SrV-A
CTC 212	*Citrus reticulata* Murcott	Metropolitana (Mallarauco)	16SrV-A
CTC 134	*Citrus reticulata* Orri	L. B. O’Higgins (Peumo)	16SrXIII-F
CTC 170	*Citrus* × *sinensis* Fukumoto	L. B. O’Higgins (Pichidegua)	16SrXIII-F

**Table 3 pathogens-11-00048-t003:** Number of citrus plants collected in the main producing regions of Chile.

Region	Number of Samples
Tarapacá	10
Coquimbo	60
Valparaíso	55
Metropolitana	62
Libertador Bernardo O’Higgins	49
Maule	22

## Data Availability

The GenBank Accession Numbers presented in this study are openly available in the National Center for Biotechnology Information (NCBI) at NCBI nucleotide (https://www.ncbi.nlm.nih.gov/nucleotide/, accessed on 15 November 2021).

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
