# Peer review of "Survey for ‘Candidatus Liberibacter’ and ‘Candidatus Phytoplasma’ in Citrus in Chile"

_pathogens, 2021, doi:10.3390/pathogens11010048_

Round 1
Reviewer 1 Report
This manuscript entitled “Survey for `Candidatus Liberibacter' and `Candidatus Phytoplasma' in citrus in Chile” was first report of phytoplasma 16SrV-A and 16SrXIII-F in citrus. However, the title of manuscript should be changed. This manuscript focused on described novel phytoplasmas strain. Very little research about Survey for CLas.
line 22, authors should delete the references, in general, no references are in abstract.
For identification of novel phytoplasma strain, detection of phytoplasma by TEM should be performed. Author need to add these data.

Reviewer 2 Report
L23; “This is the first report of these phytoplasmas in citrus worldwide.” Authors may need to justify the significance of differences of “phytoplasmas enclosed in subgroups 16SrV-A (12) and 16SrXIII-F (2)” compared to already reported Candidatus phytoplasmas species in citrus or amend the statement to tone down.
The authors may consider modifying the introduction section with appropriate original citations. For example: reference [1] should have been “Bové 2006.” The references should be included while considering the chronological order for the information reported. Please consider revising this issue throughout the manuscript.
The result section should begin with explaining the sampling. Here, the result section begins with “All the PCR analyses for the detection of 'Candidatus Liberibacter' species were negative.” But its not clear the “PCR analyses” of what kind of samples?
Please make sure, wherever is applicable, the genus and species names are italicized.
Authors need to provide unmodified original raw images for Figure 2 and Figure 3 as supplementary information, clearly displaying the loading wells.
Round 2
Reviewer 1 Report
no comments
Reviewer 2 Report
The authors have addressed my concerns appropriately.